# MProto: Multi-Prototype Network with Denoised Optimal Transport for Distantly Supervised Named Entity Recognition

**Shuhui Wu**[1] and **Yongliang Shen**[1†] and **Zeqi Tan**[1]
**Wenqi Ren**[2] and **Jietian Guo**[2] and **Shiliang Pu**[2] and **Weiming Lu**[1†]
[1]College of Computer Science and Technology, Zhejiang University, China
[2] Hikvision Research Institute
{shwu,syl,zqtan}@zju.edu.cn
{renwenqi,guojietian,pushiliang.hri}@hikvision.com
luwm@zju.edu.cn

## Abstract

Distantly supervised named entity recognition (DS-NER) aims to locate entity mentions and classify their types with only knowledge bases or gazetteers and unlabeled corpus. However, distant annotations are noisy and degrade the performance of NER models. In this paper, we propose a noise-robust prototype network named MProto for the DS-NER task. Different from previous prototype-based NER methods, MProto represents each entity type with multiple prototypes to characterize the intra-class variance among entity representations. To optimize the classifier, each token should be assigned an appropriate ground-truth prototype and we consider such token-prototype assignment as an optimal transport (OT) problem. Furthermore, to mitigate the noise from incomplete labeling, we propose a novel denoised optimal transport (DOT) algorithm. Specifically, we utilize the assignment result between Other class tokens and all prototypes to distinguish unlabeled entity tokens from true negatives. Experiments on several DS-NER benchmarks demonstrate that our MProto achieves state-of-the-art performance. The source code is now available on Github[1].

## 1 Introduction

Named Entity Recognition (NER) is a fundamental task in natural language processing and is essential in various downstream tasks (Li and Ji, 2014; Miwa and Bansal, 2016; Ganea and Hofmann, 2017; Shen et al., 2021b). Most state-of-the-art NER models are based on deep neural networks and require massive annotated data for training. However, human annotation is expensive, time-consuming, and often unavailable in some specific domains. To mitigate the reliance on human annotation, distant supervision is widely applied to automatically generate annotated data with the aid of only knowledge bases or gazetteers and unlabeled corpus.

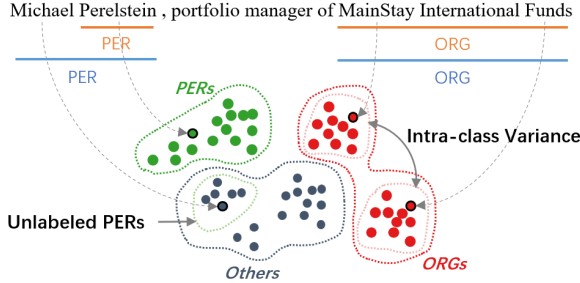

Figure 1: The distant annotations are marked in orange, and the human annotations are marked in blue. Here we illustrate two types of issues in the DS-NER. First, unlabeled entities will introduce incomplete labeling noise in O class. Second, tokens of the same class may fall into different sub-clusters due to semantic differences.

Despite its easy accessibility, the distantly-annotated data is rather noisy, severely degrading the performance of NER models. We observe two types of issues in the distantly-supervised named entity recognition task. The first issue is **incomplete labeling**. Due to the limited coverage of dictionaries, entities not presented in dictionaries are mistakenly labeled as Other class (denoted as O class). For instance, in Figure 1, "Micheal" is not covered in the dictionary and thus cannot be correctly labeled by distant supervision. These unlabeled entities will misguide NER models to overfit the label noise, particularly hurting recall. The second issue is **intra-class variance**. As illustrated in Figure 1, tokens of the same class (e.g., "MainStay" and "Funds") may fall into different sub-clusters in feature space due to semantic differences. Traditional single-prototype classifiers do not consider the semantic difference within the same entity type. They set only one prototype for each type, which suffers from intra-class variance.

Various methods have been proposed to mitigate the noise of incomplete labeling. For example, AutoNER (Shang et al., 2018) tries to modify the standard CRF-based classifier to adapt to the noisy NER datasets. Self-learning-based methods (Liang

---

† Corresponding author.

[1]https://github.com/XiPotatonium/mproto

et al., 2020; Meng et al., 2021; Zhang et al., 2021c) learn by soft labels generated from a teacher model for denoising. Positive-unlabeled learning methods treat tokens of O class as unlabeled samples and optimize with an unbiasedly estimated empirical risk (Peng et al., 2019; Zhou et al., 2022). Some other studies utilize negative sampling to avoid NER models overfitting label noise in O class (Li et al., 2021, 2022). Despite these efforts, the issue of intra-class variance has not yet been studied in previous DS-NER research.

Unlike previous methods, we propose a novel prototype-based network named MProto for distantly-supervised named entity recognition. In MProto, each class is represented by multiple prototypes so that each prototype can represent a certain sub-cluster of an entity type. In this way, our MProto can characterize the intra-class variance. To optimize our multiple-prototype classifier, we assign each input token with an appropriate ground-truth prototype. We formulate such token-prototype assignment problem as an optimal transport (OT) problem. Moreover, we specially design a denoised optimal transport (DOT) algorithm for tokens labeled as O class to alleviate the noise of incomplete labeling. Specifically, we perform the assignment between O tokens and all prototypes and regard O tokens assigned to O prototypes as true negatives while tokens assigned to prototypes of entity classes as label noise. Based on our observation, before overfitting, unlabeled entity tokens tend to be assigned to prototypes of their actual classes in our similarity-based token-prototype assignment. Therefore, unlabeled entity tokens can be discriminated from clean samples by our DOT algorithm so as not to misguide the training.

Our main contributions are three-fold:

- We present MProto for the DS-NER task. MProto represents each entity type with multiple prototypes for intra-class variance. And we model the token-prototype assignment problem as an optimal transport problem.

- To alleviate the noise of incomplete labeling, we propose the denoised optimal transport algorithm. Tokens labeled as O but assigned to non-O prototypes are considered as label noise so as not to misguide the training.

- Experiments on various datasets show that our method achieves SOTA performance. Further analysis validates the effectiveness of our

multiple-prototype classifier and denoised optimal transport.

## 2 Method

In this section, we first introduce the task formulation in Section 2.1. Then, we present our MProto network in Section 2.2. To compute the cross-entropy loss, we assign each token with a ground-truth prototype, and the token-prototype assignment will be discussed in Section 2.3. Besides, to mitigate the incomplete labeling noise in O tokens, we propose the denoised optimal transport algorithm which will be specified in Section 2.4. Figure 2 illustrates the overview of our MProto.

### 2.1 Task Formulation

Following the tagging-based named entity recognition paradigm, we denote the input sequence as $X = [x_1, \cdots, x_L]$ and the corresponding distantly-annotated tag sequence as $Y = [y_1, \ldots, y_L]$. For each token-label pair $(x_i, y_i)$, $x_i$ is the input token and $y_i \in \mathcal{C} = \{c_1, \ldots, c_K\}$ is the class of the token. Here we let $c_1$ be O class and others be predefined entity classes. We denote the human-annotated tag sequence as $\tilde{Y} = [\tilde{y}_1, \ldots, \tilde{y}_L]$. Human annotations can be considered as true labels. In the distantly-supervised NER task, only distant annotations can be used for training while human annotations are only available in evaluation.

### 2.2 Multi-Prototype Network

**Token Encoding.** For each input token sequence $X$, we generate its corresponding features with a pretrained language model:

$$\mathbf{H} = f_\theta(X) \qquad (1)$$

where $f_\theta$ is the pretrained language model parameterized by $\theta$ and $\mathbf{H} = [\mathbf{h}_1, \ldots, \mathbf{h}_L]$ is the features of the token sequence.

**Prototype-based Classification.** MProto is a prototype-based classifier where predictions are made based on token-prototype similarity. Visualization in Figure 4 shows that tokens of the same class will form multiple sub-clusters in the feature space due to the semantic difference. However, previous prototype-based classifiers which represent each entity type with only a single prototype cannot characterize such intra-class variance. To this end, we represent each entity type with multiple prototypes. Specifically, for each class $c$, let

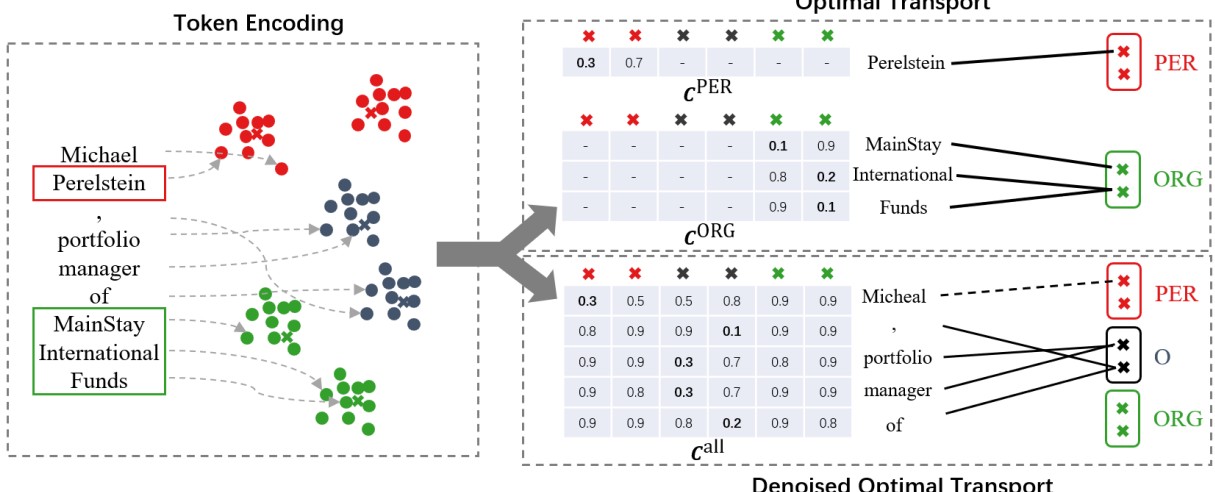

Figure 2: The overview of MProto. For the clarity, we only show tokens of three classes (PER, ORG and O) and MProto with two prototypes per class. MProto is a prototype-based classifier where predictions are made based on the similarity between token features and prototypes. To optimize the multiple-prototype classifier, we assign each token an appropriate ground-truth prototype. The top right of the figure illustrates the token-prototype assignment for entity tokens which is considered as an OT problem. The bottom right of the figure illustrates the token-prototype assignment for O tokens which is solved by our DOT algorithm where O tokens assigned with non-O prototypes (in dashed line) are considered as label noise. The assignment is done based on cost matrix (the tables shown in the figure) where each cell is the distance between a token and a prototype.

$\mathcal{P}_c = \{\mathbf{p}_{c,1}, \ldots, \mathbf{p}_{c,M}\}$ be the set of $M$ prototypes representing class $c$. For the classification, we compare the similarity between token features and all prototypes, and the class of the most similar prototype is chosen as the prediction:

$$\hat{c}_i = \arg\max_c \mathrm{s}(\mathbf{h}_i, \mathbf{p}_{c,m}) \qquad (2)$$

where $\mathrm{s}(\mathbf{h}, \mathbf{p}) = \frac{\mathbf{h} \cdot \mathbf{p}}{||\mathbf{h}|| \cdot ||\mathbf{p}||}$ is the similarity metric and here we adopt cosine similarity. In the inference process, consecutive tokens of the same type are considered a singular entity.

**Loss.** To update the parameters of our MProto, we calculate the loss between token features and their corresponding ground-truth prototypes. First, we should assign each token with an appropriate ground-truth prototype. Specifically, for each token $i$ and its annotation $y_i = c_i$, one of the prototypes of class $c_i$ will be assigned as the ground truth based on the similarity between the token feature and prototypes. Such token-prototype assignment is considered as an optimal transport problem, and the detail of solving the token-prototype assignment will be discussed later (in Section 2.3). Based on the assigned ground-truth prototype $\mathbf{p}_{c_i,m}$ for token $i$, we can compute the cross-entropy loss:

$$\ell^{\mathrm{CE}} = -\sum_i \log \frac{\exp(\mathrm{s}(\mathbf{h}_i, \mathbf{p}_{c_i,m}))}{\sum_{c',m'} \exp(\mathrm{s}(\mathbf{h}_i, \mathbf{p}_{c',m'}))} \qquad (3)$$

However, optimizing the model only through CE loss guides tokens to be **relatively** close to their ground-truth prototypes in feature space, while the compactness of the token features within the same sub-cluster is not considered. To this end, we further optimize the absolute distance between token features and ground-truth prototypes as follows:

$$\ell^{\mathrm{c}} = \sum_i \mathrm{d}^2(\mathbf{h}_i, \mathbf{p}_{c_i,m}) = \sum_i (1 - \mathrm{s}(\mathbf{h}_i, \mathbf{p}_{c_i,m}))^2 \qquad (4)$$

here we define the distance based on cosine similarity as $\mathrm{d}(\mathbf{h}_i, \mathbf{p}_{c_i,m}) = 1 - \mathrm{s}(\mathbf{h}_i, \mathbf{p}_{c_i,m})$. The overall loss can be calculated as follows:

$$\ell = \ell^{\mathrm{CE}} + \lambda^{\mathrm{c}} \ell^{\mathrm{c}} \qquad (5)$$

where $\lambda^{\mathrm{c}}$ is the weight of the compactness regularization.

**Prototype Updating.** We update each prototype with the token features assigned to that prototype. For convenience, we denote the set of tokens assigned to prototype $p_{c,m}$ as $\mathcal{T}$. At each training step $t$, we update prototypes with exponential moving average (EMA):

$$\mathbf{p}_{c,m}^t = \alpha \mathbf{p}_{c,m}^{t-1} + (1 - \alpha) \frac{\sum_{i \in \mathcal{T}} \mathbf{h}_i}{|\mathcal{T}|} \qquad (6)$$

where $\alpha$ is the EMA updating ratio. With EMA updating, the learned prototypes can be considered

as the representative of a certain sub-cluster in the feature space.

## 2.3 Token-Prototype Assignment for Entity Tokens

In our MProto, each class is represented by multiple prototypes. Therefore, how to assign a token with an appropriate ground-truth prototype is an essential issue for optimization. First, we denote the set of tokens labeled as class $c$ as $\mathcal{T}_c$. We aim to compute the assignment matrix $\gamma^c \in \mathbb{R}^{|\mathcal{T}_c| \times M}$ between $\mathcal{T}_c$ and $\mathcal{P}_c$ (prototypes of class $c$). Then the assigned prototype $p_{c_i,m}$ for token $i$ can be obtained by $m = \arg\max_j \gamma^c_{i,j}$. We consider such token-prototype assignment problem as an optimal transport problem:

$$\hat{\gamma}^c = \arg\min_{\gamma^c} \sum_{i \in \mathcal{T}_c} \sum_{j=1}^{M} \gamma^c_{i,j} \mathbf{C}^c_{i,j}, \qquad (7)$$
$$\text{s.t.} \quad \gamma^c \mathbf{1} = \mathbf{a}, \gamma^{c\top} \mathbf{1} = \mathbf{b}$$

where $C^c_{i,j} = \mathrm{d}(\mathbf{h}_i, \mathbf{p}_{c,j})$ is the cost matrix which is composed of the distances between features of the token set $\mathcal{T}_c$ and prototype set $\mathcal{P}_c$, $\mathbf{a} = \mathbf{1} \in \mathbb{R}^{|\mathcal{T}_c|}$ is the assignment constraint for each token which guarantees that each token is expected to be assigned to exactly one prototype, $\mathbf{b} \in \mathbb{R}^M$ is the assignment constraint for each prototype which prevents the model from the trivial solution where all tokens are assigned to a single prototype. The constraint can be set based on prior knowledge. However, to keep our method simple, we simply choose even distribution ($\mathbf{b} = \frac{|\mathcal{T}_c|}{M} \mathbf{1}$). The experiments also show that such simple constraint can already achieve satisfactory performance.

By optimizing Equation 7, we aim to obtain an assignment where each token is assigned a similar prototype. The token-prototype assignment problem can be solved by the sinkhorn-knopp algorithm (Cuturi, 2013), which is detailed in Appendix A.

## 2.4 Token-Prototype Assignment for O Tokens

Incomplete labeling noise is a common issue in the distantly supervised named entity recognition task. If we directly assign all O tokens to O prototypes, features of unlabeled entity tokens will be pulled closer to O prototypes and farther to prototypes of their actual entity type, which leads to overfitting. To mitigate the noise of incomplete labeling, we specially design the denoised optimal transport algorithm for O tokens. Specifically, we allow all

prototypes to participate in the assignment of O tokens. Here we denote the set of tokens labeled as O class as $\mathcal{T}_o$ and the assignment matrix between $\mathcal{T}_o$ and all prototypes as $\gamma^o \in \mathbb{R}^{|\mathcal{T}_o| \times KM}$. The denoised optimal transport is formulated as follows:

$$\hat{\gamma}^o = \arg\min_{\gamma^o} \sum_{i \in \mathcal{T}_o} \sum_{j=1}^{KM} \gamma^o_{i,j} \mathbf{C}^{\text{all}}_{i,j}, \qquad (8)$$
$$\text{s.t.} \quad \gamma^o \mathbf{1} = \mathbf{a}, \gamma^{o\top} \mathbf{1} = \mathbf{b}$$

where $\mathbf{C}^{\text{all}} = [\mathbf{C}^{c_1}, \dots, \mathbf{C}^{c_K}] \in \mathbb{R}^{|\mathcal{T}_o| \times KM}$ is the cost matrix composed of the distances between features of the token set $\mathcal{T}_o$ and all prototypes, and $[\cdot]$ is the concatenation operation. The first constraint $\mathbf{a} = \mathbf{1} \in \mathbb{R}^{|\mathcal{T}_o|}$ indicates that each token is expected to be assigned to exactly one prototype. The second constraint is formulated as:

$$\mathbf{b} = [\frac{\beta|\mathcal{T}_o|}{M}\mathbf{1}, \underbrace{\frac{(1-\beta)|\mathcal{T}_o|}{(K-1)M}\mathbf{1}, \dots, \frac{(1-\beta)|\mathcal{T}_o|}{(K-1)M}\mathbf{1}}_{K-1}]$$
$$(9)$$

where $\beta$ is the assignment ratio for O prototypes. It indicates that: (1) we expect $\beta|\mathcal{T}_o|$ tokens to be assigned to O prototypes, (2) the remaining tokens are assigned to non-O prototypes, (3) the token features are evenly assigned to prototypes of the same type.

Intuitively, before the model overfits the incomplete labeling noise, unlabeled entity tokens are similar to prototypes of their actual entity type in feature space. So these unlabeled entity tokens tend to be assigned to prototypes of their actual entity type in our similarity-base token-prototype assignment. Therefore, tokens assigned to O prototypes can be considered as true negatives while others can be considered as label noise. We then modify the CE loss in Equation 3 as follows:

$$\ell^{\text{CE}} = -\sum_{i \in \mathcal{T}_{\neg o}} \log \frac{\exp(\mathrm{s}(\mathbf{h}_i, \mathbf{p}_{c_i,m}))}{\sum_{c',m'} \exp(\mathrm{s}(\mathbf{h}_i, \mathbf{p}_{c',m'}))}$$
$$- \sum_{i \in \mathcal{T}_o} w_i \log \frac{\exp(\mathrm{s}(\mathbf{h}_i, \mathbf{p}_i))}{\sum_{c',m'} \exp(\mathrm{s}(\mathbf{h}_i, \mathbf{p}_{c',m'}))}$$
$$(10)$$

where $\mathcal{T}_{\neg o} = \bigcup_{i=2}^{M} \mathcal{T}_{c_i}$ is the set of tokens not labeled as O, $w_i = \mathbb{1}(\mathbf{p}_i \in \mathcal{P}_{c_1})$ is the indicator with value 1 when the assigned prototype $\mathbf{p}_i$ for token $i$ is of class $c_1$ (O class). The first term of Equation 10 is the loss for entity tokens which is identical to traditional CE loss. The second term is the loss for O tokens where only these true negative

samples are considered. In this way, unlabeled entity tokens will not misguide the training of our MProto, and the noise of incomplete annotation can be mitigated.

# 3 Experiments

## 3.1 Settings

**NER Dataset.** The distantly-annotated data is generated from two wildly used flat NER datasets: (1) CoNLL03 (Tjong Kim Sang, 2002) is an open-domain English NER dataset that is annotated with four entity types: PER, LOC, ORG and MISC. It is split into 14041/3250/3453 train/dev/test sentences. (2) BC5CDR (Wei et al., 2016) is a biomedical domain English NER dataset which consists of 1500 PubMed articles with 4409 annotated Chemicals and 5818 Diseases. It is split into 4560/4579/4797 train/dev/test sentences.

**Distant Annotation.** We generate four different distantly-annotated datasets based on CoNLL03 and BC5CDR for the main experiment: (1) CoNLL03 (Dict) is generated by dictionary matching with the dictionary released by Peng et al. (2019). (2) CoNLL03 (KB) is generated by KB matching follows (Liang et al., 2020). (3) BC5CDR (Big Dict) is generated by dictionary matching with the dictionary released by Shang et al. (2018). (4) BC5CDR (Small Dict) is generated by the first 20% of the dictionary used in BC5CDR (Big Dict). We follow the dictionary matching algorithm presented in (Zhou et al., 2022) and the KB matching algorithm presented in (Liang et al., 2020).

**Evaluation Metric.** We train our MProto with the distantly-annotated train set. Then the model is evaluated on the human-annotated dev set. The best checkpoint on the dev set is tested on the human-annotated test set, and the performance on the test set is reported as the final result. Entity prediction is considered correct when both span and category are correctly predicted.

**Implementation Details.** For a fair comparison, we use the BERT-base-cased (Devlin et al., 2019) for the CoNLL03 dataset, and BioBERT-base-cased (Lee et al., 2020) for the BC5CDR dataset. We set $M = 3$ prototypes per class and the hyperparameter search experiment can be found in Appendix E. More detailed hyperparameters can be found in Appendix B.

## 3.2 Baselines

**Fully-Supervised.** We implement a fully-supervised NER model for comparison. The model is composed of a BERT encoder and a linear layer as the classification head. Fully-supervised model is trained with human-annotated data, and the performance can be viewed as the upper bound of the DS-NER models.

**AutoNER.** AutoNER (Shang et al., 2018) is a DS-NER method that classifies two adjacent tokens as break or tie. To mitigate the noise of incomplete labeling, the unknown type is not considered in loss computation. For a fair comparison, we re-implement their method to use BERT-base-cased as the encoder for the CoNLL03 dataset and use BioBERT-base-cased for the BC5CDR dataset.

**Early Stoping.** Liang et al. (2020) apply early stopping to prevent the model from overfitting the label noise. For a fair comparison, we re-implement their method to use BERT-base-cased as the encoder for the CoNLL03 dataset and use BioBERT-base-cased for the BC5CDR dataset.

**Negative Sampling.** Li et al. (2021, 2022) use negative sampling to eliminate the misguidance of the incomplete labeling noise. We re-implement their method and set the negative sampling ratio to be 0.3 as recommended in (Li et al., 2021).

**MPU.** Multi-class positive-unlabeled (MPU) learning (Zhou et al., 2022) considers the negative samples as unlabeled data and optimizes with the unbiasedly estimated task risk. Conf-MPU incorporates the estimated confidence score for tokens being an entity token into the MPU risk estimator to alleviate the impact of annotation imperfection.

## 3.3 Overall Performance

Table 1 shows the overall performance of our MProto compared with various baselines in four distantly-annotated datasets. We observe that our MProto achieves state-of-the-art performance on all four datasets. Specifically, our MProto achieves +1.48%, +0.39%, +0.28%, and +0.60% improvement in F1 score on BC5CDR (Big Dict), BC5CDR (Small Dict), CoNLL03 (KB), and CoNLL03 (Dict) compared with previous state-of-the-art methods. We also notice that our MProto achieves consistent improvement on all datasets. In contrast, previous SOTA methods usually achieve promising results on some datasets while performing poorly in other

| Model | BC5CDR (Big Dict) | BC5CDR (Small Dict) | CoNLL03 (KB) | CoNLL03 (Dict) |
|---|---|---|---|---|
| BERT (Fully Sup.) | 87.45 (85.86/89.10) | | 91.30 (90.82/91.78) | |
| DS Matching | 64.32 (86.39/51.24) | 15.69 (80.02/8.70) | 71.40 (81.13/63.75) | 63.93 (93.12/48.67) |
| AutoNER | 73.02 (78.33/68.39) | 19.90 (68.34/11.64) | 64.03 (78.17/54.21) | 59.47 (81.89/46.69) |
| Early Stopping | 73.66 (80.43/67.94) | 17.21 (75.60/ 9.71) | 77.06 (84.03/71.16) | 67.74 (86.34/55.74) |
| Neg Sampling | 78.73 (79.30/78.17) | 24.25 (78.93/14.32) | 79.30 (83.16/75.78) | 74.90 (83.71/67.78) |
| MPU | 68.22 (56.50/86.05) | 73.91 (70.08/78.18) | 65.75 (58.79/74.58) | 67.65 (63.63/72.22) |
| Conf-MPU | 77.22 (69.79/86.42) | 71.85 (81.02/64.54) | 79.16 (68.58/79.75) | 81.89 (81.71/82.08) |
| MProto | **81.47** (77.53/85.84) | **74.30** (73.41/75.22) | **79.58** (79.80/79.37) | **82.49** (84.27/80.79) |

Table 1: Overall performance. The results are reported in "F1 (precision / recall)" format.

cases. We attribute this superior performance to two main factors: (1) The denoised optimal transport algorithm significantly mitigates the noise of incomplete labeling in 0 tokens by leveraging the similarity-based token-prototype assignment result, leading to a performance improvement across varying annotation quality (with differences in dictionary coverage or Distant Supervision matching algorithms). (2) Our multiple prototype classifier can characterize the intra-class variance and can better model rich semantic entity classes, which improves the robustness across diverse data domains.

### 3.4 Ablation Study

We conduct the ablation study in the following three aspects. The results are shown in Table 2.

**Multiple Prototypes.** By representing each entity type with multiple prototypes, the F1 score of MProto improves by +0.95% on BC5CDR (Big Dict) and by +1.26% on CoNLL03 (Dict). It confirms that the multiple-prototype classifier can greatly help the MProto characterize the intra-class variance of token features.

**Compactness Loss.** By applying compactness loss as elaborated in Equation 4, the F1 score of MProto improves by +2.23% on BC5CDR and by +0.29%. Simply optimizing the Cross-Entropy loss in Equation 3 only encourages the token features to be relatively close to the assigned ground-truth prototypes. While the compactness loss directly optimizes the distance between the token feature and the prototype to be small, making the token features of the same prototype more compact.

**Denoised Optimal Transport.** Compared with assigning all 0 tokens to 0 prototypes, the denoised optimal transport elaborated in Section 2.4 improves the F1 score of MProto by +21.89% on BC5CDR and by +34.14% on CoNLL03. The improvement indicates that the denoised optimal transport significantly mitigates the incomplete labeling

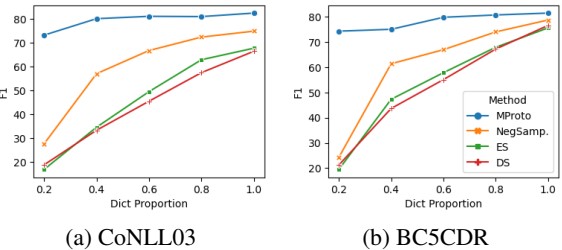

(a) CoNLL03     (b) BC5CDR

Figure 3: Results on the distantly-annotated dataset with different proportions of the dictionaries. "NegSamp." stands for Negative Sampling. "ES" stands for Early Stopping. "DS" stands for dictionary matching.

noise in the DS-NER task.

### 3.5 Experiments on Dictionary Coverage

To analyze the performance of our method with different coverage of dictionaries, we generate distant annotations with dictionaries of different proportions following (Zhou et al., 2022). The distant annotation quality can be found in Table 3 and the experiment result is shown in Figure 3.

The result shows that the performance of Negative Sampling and Early Stopping drops significantly when the coverage of the dictionary decreases. When the dictionary size drops from 100% to 20%, the F1 score of the Negative Sampling drops by -54.48% on the BC5CDR dataset and by -47.44% on the CoNLL03 dataset. The F1 score of the Early Stopping drops by -56.13% on the BC5CDR dataset and by -51.02% on the CoNLL03 dataset. This indicates that these methods suffer severely from incomplete labeling noise. However, our MProto only suffers from a slight performance drop compared with Negative Sampling and Early Stopping. The F1 score drops only by -7.17% on BC5CDR and by -9.29% on CoNLL03. We can conclude that with multiple prototypes and denoised optimal transport, our MProto is more robust to the incomplete labeling noise and can achieve better performance, especially on the DS-

| Model | BC5CDR (Big Dict) | | | | | CoNLL03 (Dict) | | | | |
|---|---|---|---|---|---|---|---|---|---|---|
| | Loc. F1 | Cls. F1 | Prec. | Rec. | F1 | Loc. F1 | Cls. F1 | Prec. | Rec. | F1 |
| Default | **81.80** | 92.21 | 77.53 | 85.84 | **81.47** | **89.04** | **86.31** | 84.27 | **80.79** | **82.49** |
| w/o Multiple Prototypes | 80.81 | **92.29** | 75.74 | **85.94** | 80.52 | 88.72 | 85.22 | **84.39** | 79.27 | 81.75 |
| w/o Compactness Loss | 79.64 | 91.05 | 75.05 | 83.92 | 79.24 | 88.79 | 86.06 | 84.09 | 80.40 | 82.20 |
| w/o DOT | 59.53 | 63.05 | **81.94** | 46.20 | 59.09 | 59.58 | 54.59 | 58.63 | 41.13 | 48.35 |

Table 2: Ablation Study.

| Dataset | Prec. | Rec. | F1 |
|---|---|---|---|
| BC5CDR (20% Dict) | 87.31 | 12.09 | 21.24 |
| BC5CDR (40% Dict) | 87.66 | 29.06 | 43.65 |
| BC5CDR (60% Dict) | 88.42 | 40.02 | 55.10 |
| BC5CDR (80% Dict) | 89.42 | 53.85 | 67.22 |
| BC5CDR (100% Dict) | 89.45 | 66.95 | 76.58 |
| CoNLL03 (20% Dict) | 81.49 | 10.55 | 18.67 |
| CoNLL03 (40% Dict) | 84.87 | 20.65 | 33.21 |
| CoNLL03 (60% Dict) | 87.58 | 30.68 | 45.44 |
| CoNLL03 (80% Dict) | 89.02 | 42.42 | 57.46 |
| CoNLL03 (100% Dict) | 91.62 | 52.15 | 66.47 |
| CoNLL03 (KB) | 82.31 | 62.17 | 70.84 |

Table 3: The distant annotation quality (span-level precision/recall/f1) of the datasets generated with different dictionaries/KBs.

NER datasets with low-coverage dictionaries.

### 3.6 Visualization of Tokens and Prototypes

We get the token features and the embedding of the prototypes from the best checkpoint on the dev set. To visualize high-dimensional features, we convert token features and prototype embeddings to 2-dimensions by t-SNE.

As illustrated in Figure 4, we can conclude that: (1) The intra-class variance is a common phenomenon in the NER task. The token features of Disease class form two sub-clusters. Representing each entity type with only a single prototype cannot cover token features in all sub-clusters. Therefore, single-prototype networks cannot characterize the intra-class variance and the performance will suffer. (2) Our MProto can model the intra-class variance in the NER task. For each sub-cluster of Disease, there exists at least one prototype to represent that sub-cluster. In this way, the performance of the NER method can benefit from the multiple-prototype network.

More visualizations on different datasets and MProto with different prototypes can be found in Appendix C.

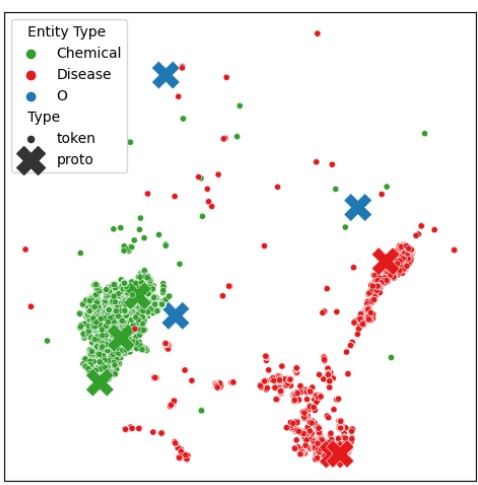

Figure 4: The t-SNE visualization of token features and prototypes on BC5CDR (Big Dict). We omit the enormous O tokens for the clarity of demonstration.

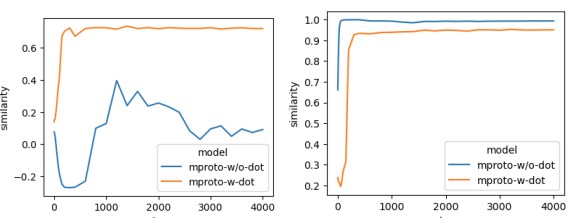

(a) Entity token-prototype similarity

(b) O token-prototype similarity

Figure 5: The cosine similarity curve between tokens and their actual prototypes over training steps

### 3.7 Effectiveness of Denoised Optimal Transport

To validate the efficacy of our DOT algorithm, we present the similarity between tokens and prototypes over training steps in Figure 5. The token-prototype similarity of each entity type is obtained by:

$$\text{sim}_c = \frac{1}{|\chi_c|} \sum_{\mathbf{x} \in \chi_c} \max_m s(\mathbf{x}, \mathbf{p}_{c,m}) \qquad (11)$$

where $\chi_c$ is the set of tokens whose actual class is

$c$ (the actual class of token is obtained from human annotation of CoNLL03) and $s$ denotes the cosine similarity function. As can be observed from the figure, when the model is trained using the DOT, the similarity between entity tokens and their actual prototypes exhibits a consistent increase throughout the training, thereby bolstering the entity token classification performance. In contrast, the model trained without DOT exhibits a decrease in similarity, suggesting that the model overfits the incomplete labeling noise. Regarding O tokens, the similarity between O tokens and O prototypes is higher when trained without DOT. This implies the network tends to predict the O class, which leads to false negative. While with our DOT algorithm, this issue can be mitigated. Based on these observations, we can conclude that the DOT algorithm plays a important role in alleviating the noise of incomplete labeling, and thereby significantly enhancing the performance in the DS-NER task.

## 4 Related Work

Named entity recognition is a fundamental task in natural language processing and has been applied to various downstream tasks (Li and Ji, 2014; Miwa and Bansal, 2016; Ganea and Hofmann, 2017; Wu et al., 2020; Shen et al., 2021b; Wu et al., 2022b, 2023). NER methods can be divided into two main categories: tagging-based and span-based. Tagging-based methods (Lample et al., 2016; Huang et al., 2015) predict a label for each token, which perform well at detecting flat named entities while failing at detecting nested named entities. Span-based methods (Sohrab and Miwa, 2018; Yu et al., 2020; Wang et al., 2020; Shen et al., 2021a) perform classification over span representations, which performs well on the nested NER task but is inferior in computational complexity. Tan et al. (2021); Shen et al. (2022); Wu et al. (2022a) design a query-based NER framework that optimizes entity queries using bipartite graph matching. Recently, some generative NER methods (Yan et al., 2021; Shen et al., 2023a,b; Lu et al., 2022) have been proposed with superior performance on various NER tasks. These supervised NER methods require a large amount of annotated data for training.

**DS-NER.** To mitigate the need for human annotations, distant supervision is widely used. The main challenge of the DS-NER task is the label noise, of which the most widely studied is the incomplete labeling noise. Various methods have been proposed to address the noise in distant annotations. AutoNER (Shang et al., 2018) design a new tagging scheme that classifies two adjacent tokens to be tied, broken, or unknown. Token pairs labeled unknown are not considered in loss computation for denoising. Negative sampling methods (Li et al., 2021, 2022) sample O tokens for training to mitigate the incomplete labeling noise. Positive-unlabeled learning (PU-learning) (Peng et al., 2019; Zhou et al., 2022) treats tokens labeled with O class as unlabeled samples and optimizes with an unbiasedly estimated empirical risk. Self-learning-based methods (Liang et al., 2020; Zhang et al., 2021c; Meng et al., 2021) train a teacher model with distant annotations and utilize the soft labels generated by the teacher to train a new student. Other studies also adopt causal intervention (Zhang et al., 2021a) or hyper-geometric distribution (Zhang et al., 2021b) for denoising. In our MProto, we propose a novel denoised optimal transport algorithm for the DS-NER task. Experiments show that the denoised optimal transport can significantly mitigate the noise of incomplete labeling.

**Prototypical Network.** Our work is also related to prototypical networks (Snell et al., 2017). Prototypical networks learn prototype for each class and make predictions based on the similarity between samples and prototypes. It is widely used in many tasks such as relation extraction (Ding et al., 2021) and the few-shot named entity recognition (FS-NER) task (Fritzler et al., 2019; Ma et al., 2022). Previous prototypical networks for the NER task are mostly single-prototype networks, which do not consider the semantic difference within the same entity type. Tong et al. (2021) try to incorporate multiple prototypes for O class in the FS-NER task. However, they do not consider the intra-class variance in entity classes and their clustering strategy will introduce huge computational complexity when training data is large. In our work, we utilize multiple prototypes to represent each entity type and solve the token-prototype assignment efficiently with the sinkhorn-knopp algorithm. Experiments confirm that our MProto can successfully describe the intra-class variance.

## 5 Conclusion

In this paper, we present MProto, a prototype-based network for the DS-NER task. In MProto, each category is represented with multiple prototypes to

model the intra-class variance. We consider the token-prototype assignment problem as an optimal transport problem and we apply the sinkhorn-knopp algorithm to solve the OT problem. Besides, to mitigate the noise of incomplete labeling, we propose a novel denoised optimal transport algorithm for O tokens. Experiments show that our MProto has achieved SOTA performance on various DS-NER benchmarks. Visualizations and detailed analysis have confirmed that our MProto can successfully characterize the intra-class variance, and the denoised optimal transport can mitigate the noise of incomplete labeling.

## Limitations

The intra-class variance is actually a common phenomenon in the NER task. Currently, we only try to utilize the MProto on the distantly supervised NER task. Applying MProto to the few-shot NER task and the fully supervised NER task can be further investigated.

## Acknowledgements

This work is supported by the Fundamental Research Funds for the Central Universities (No. 226-2023-00060), National Natural Science Foundation of China (No. 62376245), Key Research and Development Program of Zhejiang Province (No. 2023C01152), National Key Research and Development Project of China (No. 2018AAA0101900), Joint Project DH-2022ZY0013 from Donghai Lab, and MOE Engineering Research Center of Digital Library.

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

## A  Sinkhorn-Knopp Algorithm

We apply the sinkhorn-knopp algorithm (Cuturi, 2013) to solve the optimal transport and denoised optimal transport elaborated in equation 7 and equation 8. In order to apply the sinkhorn-knopp algorithm, we add an entropy regularizer as follows:

$$\hat{\gamma} = \arg\min_{\gamma} \sum_i \sum_j \gamma_{i,j} \mathbf{C}_{i,j} + \lambda^r \, \mathrm{H}(\gamma), \tag{12}$$
$$\text{s.t.} \quad \gamma \mathbf{1} = \mathbf{a}, \gamma^\top \mathbf{1} = \mathbf{b}$$

where $\lambda^r$ is the weight of the regularization and $\mathrm{H}(\gamma) = \sum_{i,j} \gamma_{i,j} \log(\gamma_{i,j})$ is the entropy of the assignment matrix.

We specify the pseudo-code for the sinkhorn-knopp algorithm in Algorithm 1. Here the $\oslash$ corresponds to the element-wise division, $\mathbf{a}$ and $\mathbf{b}$ are vectors that represent the weights of each sample in the source and target distributions, $\mathbf{C}$ is the cost matrix, $\lambda^r$ is the regularization weight, and $\gamma$ is the assignment matrix.

The sinkhorn-knopp algorithm mainly consists of several matrix operations which can be easily accelerated by GPU devices. And we empirically find that the sinkhorn-knopp algorithm can obtain satisfying results in a few iterations in our work. Therefore, applying the sinkhorn-knopp algorithm to solve the token-prototype assignment problem only has a slight impact on the speed of the training.

---

**Algorithm 1** Sinkhorn-Knopp Algorithm

---

**Require:** $\mathbf{a}, \mathbf{b}, \mathbf{C}, \lambda^r$
  $\mathbf{u}^0 = \mathbf{1}, \mathbf{K} = \exp(-\mathbf{C}/\lambda^r)$
  **for** $i$ in $1, \ldots, n$ **do**
    $\mathbf{v}^i = \mathbf{b} \oslash \mathbf{K}^\top \mathbf{u}^{i-1}$
    $\mathbf{u}^i = \mathbf{a} \oslash \mathbf{K} \mathbf{v}^i$
  **end for**
  **return** $\gamma = \mathrm{diag}(\mathbf{u}^n) \mathbf{K} \, \mathrm{diag}(\mathbf{v}^n)$

---

## B  Implementation Details and Hyperparameters

We implement our MProto with Pytorch[2] and Huggingface Transformers[3]. All experiments are carried out on a single RTX-3090 with 24G graphical memory. And the training of the model can be finished in approximately 2 hours.

We report the hyperparameters used in different datasets in Table 4. When training, we use an

---

https://pytorch.org/
[3] https://huggingface.co/transformers

|  | BC5CDR | | CoNLL03 | |
|---|---|---|---|---|
|  | Big Dict | Small Dict | KB | Dict |
| $M$ | 3 | 3 | 3 | 3 |
| $\lambda^c$ | 0.05 | 0.1 | 0.05 | 0.01 |
| $\alpha$ | 0.9 | 0.5 | 0.5 | 0.9 |
| $\beta$ | 0.01 | 0.01 | 0.05 | 0.01 |

Table 4: Hyperparameters used in different datasets.

AdamW optimizer with weight decay 0.0001 and maximum gradient norm 1.0. The maximum learning is 0.0001, and the learning rate is warmed up linearly in the first 100 steps and decayed linearly afterward. The batch size is set to 32. We train the model for 10 epochs in all experiments.

For the sinkhorn-knopp algorithm used in computing the token-prototype assignment, we set the regularization weight $\lambda^r = 0.001$ and the number of iterations to 100.

## C  Feature Visualization

We further visualize the token features and the embedding of the prototypes in different datasets and model setups in Figure 6.

As shown in Figure 6a and Figure 6c, we can see that even when the model is trained with only a single prototype per entity type ($M = 1$), token features of the same entity type still tend to form different sub-clusters due to the semantic difference. Therefore, it can be confirmed that the intra-class variance is a common phenomenon in the NER task regardless of the model.

Our MProto represents each entity type with multiple prototypes. As shown in Figure 4 and Figure 6b, for each sub-cluster of token features, there exists at least one prototype to represent this sub-cluster. In other words, our MProto can successfully model the intra-class variance of the entity token features. The visualizations show that representing each entity type with multiple prototypes rather than a single prototype is beneficial and can significantly improve the performance in the DS-NER task.

## D  Transport Plan of Unlabeled Entities

To analyze the effectiveness of our denoised optimal transport algorithm, we obtain the transport plan by counting the assignment result of the unlabeled entity tokens in the train set with a checkpoint of MProto at the early stage of training.

As shown in Figure 7, most unlabeled entity

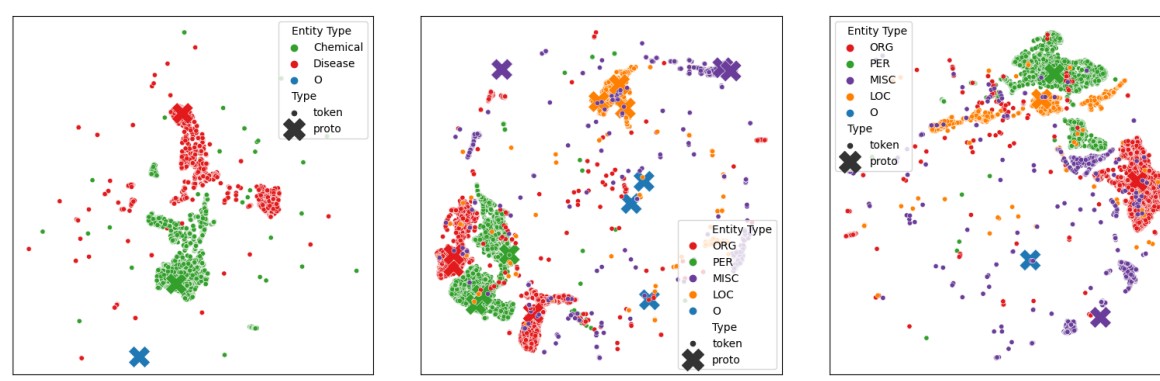

(a) MProto ($M = 1$) on BC5CDR (Big Dict)  (b) MProto ($M = 3$) on CoNLL03 (Dict)  (c) MProto ($M = 1$) on CoNLL03 (Dict)

Figure 6: The t-SNE visualization of token features and prototypes. The visualization for MProto ($M = 3$) on BC5CDR (Big Dict) can be found in Figure 4.

| $M$ | BC5CDR (Big Dict) | | | CoNLL03 (Dict) | | |
|---|---|---|---|---|---|---|
| | Prec. | Rec. | F1 | Prec. | Rec. | F1 |
| 1 | 75.74 | **85.94** | 80.52 | 84.39 | 79.27 | 81.75 |
| 2 | 77.36 | 83.43 | 80.28 | 83.42 | 79.27 | 81.29 |
| 3 | **77.53** | 85.84 | **81.47** | 84.27 | **80.79** | **82.49** |
| 4 | 76.78 | 84.71 | 80.55 | **84.53** | 80.10 | 82.25 |
| 5 | 73.52 | 84.14 | 78.47 | 82.84 | 77.96 | 80.32 |
| 6 | 76.09 | 85.49 | 80.52 | 84.53 | 79.55 | 81.97 |

Table 5: Experiment with the different number of prototypes.

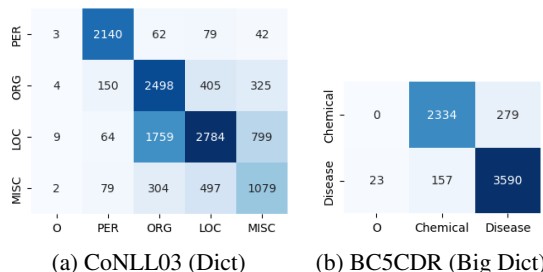

(a) CoNLL03 (Dict)  (b) BC5CDR (Big Dict)

Figure 7: The transport plan for unlabeled entities. The y-axis represents the actual entity type of the unlabeled entity tokens. The x-axis represents the class of the prototypes that are assigned to the unlabeled entity tokens.

tokens are assigned to prototypes of their actual classes, and only a few tokens are mistakenly assigned to the O prototypes. With this observation, we can confirm that before the model overfits the label noise, unlabeled entity tokens tend to be similar to prototypes of their actual classes in the feature space. Therefore, they tend to be assigned to prototypes of their actual classes in our similarity-based token-prototype assignment. In our denoised optimal transport, O tokens assigned to entity prototypes are considered as label noise, and only O tokens assigned to O prototypes are considered as true negative samples. In this way, we can discriminate unlabeled entity tokens from clean samples. These unlabeled entity tokens are ignored in loss computation so as not to misguide the training of the model. And we can conclude that the denoised optimal transport can effectively mitigate the incomplete labeling noise in the DS-NER task.

# E  Experiment on Different Number of Prototypes

We try different $M$ (the number of prototypes per type) on BC5CDR (Big Dict) and CoNLL03 (Dict) datasets. The results are reported in Table 5. It shows that representing each entity type with 3 prototypes is the optimal choice on both BC5CDR (Big Dict) and CoNLL03 (Dict) datasets. Intuitively, there exists an ideal $M$ for each entity type based on the semantic complexity of that category. And a too-large $M$ or a too-small $M$ will hurt the performance of MProto. Besides, setting a too-large $M$ will reduce the number of tokens assigned to each prototype. In this case, there might be no enough token features for learning representative prototypes, which leads to underfitting.

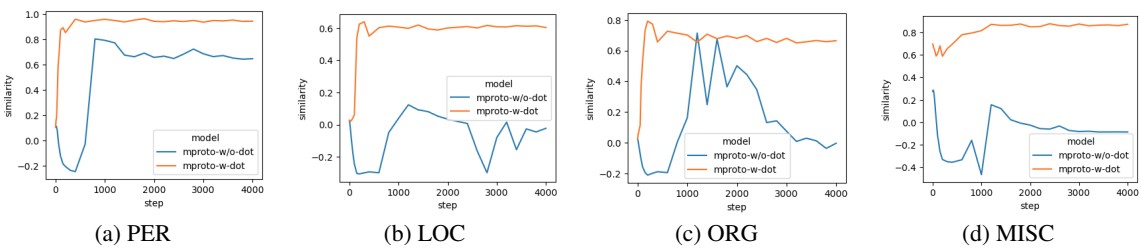

|     |     |     |     |
| :-: | :-: | :-: | :-: |
| (a) PER | (b) LOC | (c) ORG | (d) MISC |

Figure 8: The token-prototype similarity curve of each entity type over training steps.

## F  Similarity Curve of Different Entity Types

To better analyze the effectiveness of denoised optimal transport algorithm, we additionally report the similarity curve of each entity class in Figure 8. The similarity is obtained by Equation 11.