# OpenReview forum: "MProto: Multi-Prototype Network with Denoised Optimal Transport for Distantly Supervised Named Entity Recognition"
_EMNLP/2023/Conference — EMNLP 2023 Main_

### Official Review · Reviewer_orz4 · 2023-08-02

**Soundness:** 4

**Excitement:**

4: Strong: This paper deepens the understanding of some phenomenon or lowers the barriers to an existing research direction.

**Paper Topic And Main Contributions:**

The paper presents MProto, a noise-robust prototype network for distantly supervised Named Entity Recognition (DS-NER).

They use dictionary or knowledge base lookups for distantly labeling raw text with NER labels. Different to other prototypical networks they allow multiple prototypes per NER class (e.g. three "types" of ORG prototypes) to account for intra-class variance.
The prototype assignment is done via optimal transport (OT) algorithms.

Being aware that the distant labels introduce a high rate of label errors or incompleteness (due to missing entries or other), they propose a novel "denoised optimal transport" (DOT) algorithm to distinguish the O-assignes entitiey between falsely unlabeled entity tokens and true negative examples to mitigate the noise problem and avoid overfitting to the O-class.

They test their approach on two NER datasets: CoNLL03 and BC5CDR, replacing the real human labels with the distant ones from Dictionary of Knowledge Base for training. They compare their MProto approach with the fully-supervised BERT model (trained on the obviously better real human labels) as upper bound, as well as several other DS-NER models. Though naturally weaker than the baseline, MProto beats all the other DS-NER models in their experiments.

The authors present some ablation studies, e.g. only allowing one prototype per NER class, or removing the DOT handling of O-labeled tokens, the latter showing that this is indeed a very important feature of their approach. This is also supported by a comparison of the learned class prototypes and tokens that belong to the classes (in the real fully human annotated versions of the datasets).
They further notice that dictionary or knowledge base coverage is an important factor for model performance through artificially reducing coverage, which of course is a downside. However, they show that their MProto still has less performance drops compared to the other DS-NER models in comparison.



**Reasons To Accept:**

The paper very elegantly deals with two relevant problems - learning from unlabeled data via distant labels as well as mitigating label noise. It is an interesting read for both perspectives, could inspire similar approaches and overall is well written and presented. The argumentation is clear and the main idea comes across quickly. The experiments are well fitted and seem sound, as well as the ablation and secondary analyses which justify their main ideas and choices. I also appreciate that they present the drop when reducing the coverage.
The presentation (i.e. chose of sections, figures etc.) is very good and clear, also the figures in the appendix concerning token-prototype similarity are really interesting.

**Reasons To Reject:**

I do not find major reasons to reject.
The section about the (denoised) optimal transport algorithm for dealing with unlabeled entities might be a bit hard to follow or too intricate for a reader that is unfamiliar with the matter. Of course, this is not a bad thing but rather nice to learn something new. But it might help to put a more high-level or intuitive summary at the start of the section.

**Reproducibility:**

3: Could reproduce the results with some difficulty. The settings of parameters are underspecified or subjectively determined; the training/evaluation data are not widely available.

**Reviewer Confidence:**

3: Pretty sure, but there's a chance I missed something. Although I have a good feel for this area in general, I did not carefully check the paper's details, e.g., the math, experimental design, or novelty.

**Typos Grammar Style And Presentation Improvements:**

One thing that (for me) was not that clear is if the approach is token or span based. Maybe this could be made more clear?
Also, naming the source of distant labels (dictionary, knowledge base) came too late in the paper, which left me a bit confused for too long. Maybe this could be moved more to the beginning?

---

> ### Author Rebuttal · Authors · 2023-08-29
>
> Thanks for your valuable comments. Below are our responses to your concerns:
>
> ***R1: Lack of high-level or intuitive summary about the (denoised) optimal transport algorithm for dealing with unlabeled entities.***
>
> Thank you for your constructive feedback. In response to your suggestion, we will add a clearer and more accessible introduction to this section that captures the essence of the DOT method, before diving into the specifics. Here is a revised high-level summary:
>
> The denoised optimal transport (DOT) algorithm serves as a key component in our approach for dealing with unlabeled entities in Distantly Supervised Named Entity Recognition (DS-NER). Intuitively, the DOT algorithm identify and isolate unlabeled entities that are likely to be label noise. This is achieved by examining the similarity between 'Other' tokens and entity prototypes, and then assigning the tokens to their closest prototype. Only 'Other' tokens assigned to 'Other' prototypes are considered true negatives. By doing so, the algorithm can effectively distinguish between genuine unlabeled entities and label noise, ultimately reducing the risk of overfitting and improving DS-NER performance.
>
> ***Q1: If the approach is token or span based?***
>
> We apologize for the confusion. Our method is indeed a tagging-based approach that predicts a tag for each token in the text. In this process, the model assigns entity types to individual tokens, and consecutive tokens of the same type are then considered as a single entity.
>
> We appreciate your valuable suggestion and will enhance the clarity of the paper by providing more explicit details. Additionally, we plan to incorporate the source of distant labels such as dictionaries or knowledge bases earlier in the paper for the benefit of the readers.

---

### Official Review · Reviewer_7sbu · 2023-08-06

**Soundness:** 4

**Excitement:**

4: Strong: This paper deepens the understanding of some phenomenon or lowers the barriers to an existing research direction.

**Paper Topic And Main Contributions:**

This work focuses on distantly supervised named entity recognition (DS-NER) task. Distant supervision eliminates the expensive human annotation process but at the cost of noisy annotation. This paper proposes a prototype-based network approach for the DS-NER. The proposed method, referred to as MProto, considers multiple prototypes for each entity class. The multiple prototypes for an entity type help MProto to characterize sub-clusters within each entity type. MProto assigns each input token to one of the prototypes. The main contributions of this work are the formulation of token-prototype assignments. In the proposed formulation, this work also takes care of the issue of incomplete labeling, where entities are labeled as Other due to incomplete information in the knowledge resources used for distant supervision. Towards this, the two primary contributions are -
1. Formulation of token-prototype assignment as an optimal transport problem
2. To take care of the noise of incomplete labeling, a proposal of denoised optimal transport

Extensive sets of experiments and analyses are performed to establish the effectiveness of the MProto.

**Questions For The Authors:**

A few questions
1. How MProto handles label dependency between two or more consecutive tokens? In other words, how MProto handles labeling of entity mention spanning over more than one token?

2. How the complexity of the model increases with fine-grained DS-NER

**Reasons To Accept:**

1. Addresses a very challenging issue of noise and intra-class variance in the context of distantly supervised NER task.
2. The authors attempted to reduce noise arising from false negative Other (O) tagged entities with the help of multiple prototype network and denoised optimal transport method.
3. With suitable experiments, authors have established that their methods yield competitive performance against SOTA in general and domain-specific datasets.

**Reasons To Reject:**

Not very strong reasons to reject it. However, a few points
1. Limited to only two datasets with only a few numbers of entity types. Noise and corresponding challenges increase multitude with more entity types. This happens primarily due to complex entity boundaries
2. No error analysis was presented. Though authors have shown quantitative results with and without DOT, it would be interesting to see how DOT impacts assignments. Similarly for with and without multiple prototypes


**Reproducibility:**

4: Could mostly reproduce the results, but there may be some variation because of sample variance or minor variations in their interpretation of the protocol or method.

**Reviewer Confidence:**

3: Pretty sure, but there's a chance I missed something. Although I have a good feel for this area in general, I did not carefully check the paper's details, e.g., the math, experimental design, or novelty.

---

> ### Author Rebuttal · Authors · 2023-08-29
>
> Thanks for your valuable comments. Below are our responses to your concerns:
>
> ***R1:Limited to only two datasets with only a few numbers of entity types. Noise and corresponding challenges increase multitude with more entity types. This happens primarily due to complex entity boundaries.***
>
> We understand the concern regarding the limitation of using only two datasets with a limited number of entity types in our current study. However, it is worth noting that the MProto has the inherent ability to handle complex entity boundaries by representing each entity type with multiple prototypes. This enables the network to capture the diverse semantics of entities effectively, even in cases where the entity boundaries are intricate and challenging.
>
> As demonstrated in our visualization in Figure 6, the MProto's ability to represent each entity type through multiple prototypes plays a crucial role in addressing complex entity boundaries, further indicating its potential to adapt to datasets with numerous entity types.
>
> Moreover, the ablation study presented in Table 2 outlines the MProto's superior discriminative ability compared to a single-prototype counterpart. This demonstrates its adaptability and capacity to handle increased complexities and noise in entity boundaries. We believe that theoretically, MProto can maintain and enhance its performance even as the entity boundaries become more complex.
>
> We appreciate your feedback and will include more datasets with diverse entity types for a more thorough analysis in the revised version.
>
> ***R2: How DOT impacts assignments and how with and without multiple prototypes impacts assignments?***
>
> Thanks for your suggestion. To clarify the impact of DOT and multiple prototypes on assignments, we provide the assignment results of different models for unlabeled entities of BC5CDR at early stage of training in the following tables. By comparing Table1 (MProto with DOT), Table2 (MProto without DOT), and Table3 (single-prototype with DOT), we can explore the effects of both factors on the assignment process. Based on these results, we can draw two main conclusions to address the concerns raised by the reviewer:
>
> | Actual type of unlabeled tokens \\ Type of prototypes | Other | Chemical | Disease |
> | --- | --- | --- | --- |
> | Disease    | 0 | 2334 | 279 |
> | Chemical   | 23 | 157 | 3590 |
>
> *Table 1: Token-prototype assignment for MProto (M=3). Each row in these tables represents the assignment results for unlabeled tokens belonging to a specific actual entity type. Each column reflects the number of unlabeled tokens assigned to prototypes of a particular type.*
>
>
> | Actual type of unlabeled tokens \\ Type of prototypes | Other | Chemical | Disease |
> | --- | --- | --- | --- |
> | Disease    | 2613 | 0 | 0 |
> | Chemical   | 3770 | 0 | 0 |
>
> *Table 2: Token-prototype assignment for MProto (w/o DOT).*
>
> | Actual type of unlabeled tokens \\ Type of prototypes | Other | Chemical | Disease |
> | --- | --- | --- | --- |
> | Disease    | 0 | 2102 | 511 |
> | Chemical   | 32 | 457 | 3281 |
>
> *Table 3: Token-prototype assignment for MProto (M=1).*
>
> + **DOT Impact on Assignments:** We find that the utilization of DOT in the model (comparing Table 1 and Table 2) significantly improves the assignment results. In the model without DOT, all unlabeled tokens are assigned to prototypes of the "Other" type, which indicates that the model quickly overfits to label noise. In contrast, the model with DOT demonstrates a strong capacity to distinguish between unlabeled entities and 'Other' tokens, showcasing the importance of incorporating this component in the model.
>
> + **Multiple Prototype Impact on Assignments:** Comparing the results of the single-prototype network (Table 3) to the multi-prototype network (Table 1), we find that the multi-prototype model has better discriminative ability than its single-prototype counterpart. While the single-prototype network is still able to utilize the DOT algorithm effectively, the proportion of misclassifications (i.e., assigning "Chemical" tokens to "Disease" prototypes and vice versa) increases in the single-prototype network, indicating that having multiple prototypes is beneficial for the model's performance.
>
> In summary, the error analysis presented in the tables highlights the importance of both DOT and multiple prototypes in improving the assignment process. DOT effectively reduces the impact of label noise, enabling a better classification of unlabeled tokens, while multiple prototypes enhance the model's discriminative ability, resulting in a lower misclassification rate.
>
> ***Q1: How MProto handles label dependency between two or more consecutive tokens?***
>
> The tagging scheme of MProto can be viewed as IO tagging scheme (In-Out tagging scheme), that is, consecutive tokens of the same type are considered a singular entity.
>
> ***Q2: How the complexity of the model increases with fine-grained DS-NER?***
>
> In the context of fine-grained DS-NER, it may seem that increasing the granularity of entities would inherently increase the complexity of the model. However, our MProto provides an adaptive solution that mitigates this expected increase in complexity. Specifically, in the realm of fine-grained DS-NER, the nuanced granularity of entities tends to naturally form fewer clusters. This characteristic allows us to reduce the number of prototypes for each entity type by adjusting the hyperparameter $M$. By carefully setting this hyperparameter in accordance with the entity granularity, our MProto can effectively manage the potential increase in complexity that might arise with fine-grained DS-NER. In essence, the adaptability of the model's hyperparameter settings serves as a counterbalance to the increased granularity of entities, ensuring that the overall complexity of the model remains manageable.

---

### Official Review · Reviewer_vkaK · 2023-08-06

**Soundness:** 3

**Excitement:**

3: Ambivalent: It has merits (e.g., it reports state-of-the-art results, the idea is nice), but there are key weaknesses (e.g., it describes incremental work), and it can significantly benefit from another round of revision. However, I won't object to accepting it if my co-reviewers champion it.

**Paper Topic And Main Contributions:**

This paper describes a method to perform distantly supervised named-entity recognition (DS-NER), called MProto, that represent more than one prototype for each NER class. In this way, variance inside a single class can be intercepted and represented.
Results presented in the paper seem to demonstrate the effectiveness of the idea.

**Questions For The Authors:**

Why did not you compare your results with the NCBI-Disease dataset, very similar to BC5CDR?

**Reasons To Accept:**

The paper is clear and well written. The procedure is easy to follow.

**Reasons To Reject:**

There are some inconsistencies between the data presented and the previous work.
For instance, the authors use BC5CDR with dictionary in (Shang et al. 2018) and compare their work with AutoNER, described right in (Shang et al. 2018). By looking at the original paper, F1 on BC5CDR is 84.80, but in this paper is 79.99 (MProto's F1 is 81.47).
Similarly, the paper is compared with (Zhou et al. 2022), but only in its worst configuration (Conf-MPU_BERT) and not with the best one (Conf-MPU_LBiLSTM), that performs better than MProto.

**Reproducibility:**

4: Could mostly reproduce the results, but there may be some variation because of sample variance or minor variations in their interpretation of the protocol or method.

**Reviewer Confidence:**

3: Pretty sure, but there's a chance I missed something. Although I have a good feel for this area in general, I did not carefully check the paper's details, e.g., the math, experimental design, or novelty.

---

> ### Author Rebuttal · Authors · 2023-08-29
>
> Thanks for your valuable comments. Below are our responses to your concerns:
>
> ***R1: Inconsistencies between the data presented and the previous work.***
>
> We appreciate the careful examination of our work and would like to address the points you raised:
>
> + **The differences in AutoNER F1 scores:** To ensure a fair and consistent comparison across all models, we maintained a uniform encoder and chose BERT as the encoder for both our method and all the baselines. We re-implemented the AutoNER method using the BERT encoder, and the F1 score was 79.99. This is in contrast to the 84.80 originally reported by (Shang et al. 2018) using BiLSTM. This deviation can be attributed to the difference in encoders and not to the inherent capability of the model. We will provide further details on this matter in our revised version.
>
> + **The choice of Conf-MPU-BERT over Conf-MPU-LBiLSTM:** The rationale behind our choice was to maintain uniformity in encoder selection across all evaluated models. As previously stated, all methods were evaluated using the BERT encoder to ensure a consistent basis for comparison. Thus, we chose Conf-MPU-BERT over Conf-MPU-LBiLSTM as our baseline. We apologize for any confusion, and we will clarify this point in the revised paper.
>
> ***Q1: Why not compare the results with the NCBI-Disease dataset***
>
> Thank you for your insightful question regarding the choice of dataset. While both the NCBI-Disease dataset and the BC5CDR dataset are aimed at NER task within the medical domain, there are some considerations that guided our choice:
>
> + Our initial focus was to demonstrate the capabilities of MProto in handling distant-supervised NER tasks within a specific domain. The BC5CDR dataset offers not just annotations for disease mentions, but also extends to chemical mentions. This broader spectrum of annotations allowed us to demonstrate the comprehensive capabilities of MProto in handling distantly-supervised NER tasks within the medical domain.
> + BC5CDR has been commonly used in many recent studies [1; 2] on distant-supervised NER, offering a benchmark for straightforward comparison with other methods.
>
> However, we acknowledge the value of including the NCBI-Disease dataset in our evaluation and are actively working in the process of integrating this distantly-annotated dataset for evaluation. Given the limited set of directly comparable results from existing research, it will take some time in preparing the data and performing experiments on MProto and relevant baselines. We will include the results as soon as they are available.
>
> We believe that including results from various datasets strengthens the robustness of our method, and we appreciate your suggestion to include the NCBI-Disease dataset.
>
> [1] Jingbo Shang, Liyuan Liu, Xiaotao Gu, Xiang Ren, Teng Ren, and Jiawei Han. 2018. Learning named entity tagger using domain-specific dictionary. In EMNLP.
>
> [2] Kang Zhou, Yuepei Li, and Qi Li. 2022. Distantly supervised named entity recognition via confidence-based multi-class positive and unlabeled learning. In ACL.

---

### Meta-Review · Area_Chair_BsY6 · 2023-09-15

**Recommendation:** 4

**Metareview:**

The reviewers generally agreed that the paper presents a useful model for distantly supervised NER. The proposed prototype-based network, which incorporates multiple prototypes for each entity class, was praised for its ability to capture intra-class variance. The paper also introduces a denoised optimal transport algorithm to address issues of label noise.
The reviewers acknowledged the clear presentation, solid experiments, and extensive analyses performed in the paper.

However, there were some criticisms and areas for improvement highlighted by the reviewers. One reviewer pointed out inconsistencies in the presented data and comparisons to previous work, potentially caused by the differences in encoders (via author rebuttal). Nevertheless, this should be discussed and presented in the paper properly (e.g., does the proposed method still achieve SOTA performance?) Another reviewer suggested comparing the results with more datasets and configurations to further validate the approach. There was also a call for error analysis and a high-level summary of the denoised optimal transport algorithm to aid readers unfamiliar with the topic.

---

### Decision · Program_Chairs · 2023-10-07

**Decision:**

Accept-Main

**Comment:**

The reviewers generally agreed that the paper presents a useful model for distantly supervised NER. The proposed prototype-based network, which incorporates multiple prototypes for each entity class, was praised for its ability to capture intra-class variance. The paper also introduces a denoised optimal transport algorithm to address issues of label noise.
The reviewers acknowledged the clear presentation, solid experiments, and extensive analyses performed in the paper.

However, there were some criticisms and areas for improvement highlighted by the reviewers. One reviewer pointed out inconsistencies in the presented data and comparisons to previous work, potentially caused by the differences in encoders (via author rebuttal). Nevertheless, this should be discussed and presented in the paper properly (e.g., does the proposed method still achieve SOTA performance?) Another reviewer suggested comparing the results with more datasets and configurations to further validate the approach. There was also a call for error analysis and a high-level summary of the denoised optimal transport algorithm to aid readers unfamiliar with the topic.